# No Causal Effects Detected in COVID-19 and Myalgic Encephalomyelitis/Chronic Fatigue Syndrome: A Two Sample Mendelian Randomization Study

**DOI:** 10.3390/ijerph20032437

**Published:** 2023-01-30

**Authors:** Wangzi Xu, Yu Cao, Lin Wu

**Affiliations:** 1School of Public Health, Xiamen University, Xiamen 361000, China; 2Department of Clinical Laboratory, Xiang’an Hospital of Xiamen University, School of Medicine, Xiamen University, Xiamen 361000, China

**Keywords:** causal relationship, Mendelian Randomization (MR), epidemiology, Myalgic Encephalomyelitis/Chronic Fatigue Syndrome (ME/CFS), COVID-19

## Abstract

New clinical observational studies suggest that Myalgic Encephalomyelitis/Chronic Fatigue Syndrome (ME/CFS) is a sequela of COVID-19 infection, but whether there is an exact causal relationship between COVID-19 and ME/CFS remains to be verified. To investigate whether infection with COVID-19 actually causes ME/CFS, this paper obtained pooled data from the Genome Wide Association Study (GWAS) and analyzed the relationship between COVID susceptibility, hospitalization and severity of COVID and ME/CFS, respectively, using two-sample Mendelian randomization (TSMR). TSMR analysis was performed by inverse variance weighting (IVW), weighted median method, MR-Egger regression and weighted mode and simple mode methods, respectively, and then the causal relationship between COVID-19 and ME/CFS was further evaluated by odds ratio (OR). Eventually, we found that COVID-19 severity, hospitalization and susceptibility were all not significantly correlated with ME/CFS (OR:1.000,1.000,1.000; 95% CI:0.999–1.000, 0.999–1.001, 0.998–1.002; *p* = 0.333, 0.862, 0.998, respectively). We found the results to be reliable after sensitivity analysis. These results suggested that SARS-CoV-2 infection may not significantly contribute to the elevated risk of developing CFS, and therefore ME/CFS may not be a sequela of COVID-19, but may simply present with symptoms similar to those of CFS after COVID-19 infection, and thus should be judged and differentiated by physicians when diagnosing and treating the disease in clinical practice.

## 1. Introduction

An acute infectious illness caused by the severe acute respiratory syndrome coronavirus 2 (SARS-CoV-2) was first detected in December 2019 [1]. This disease, known as coronavirus disease 2019 (COVID-19), swiftly spread over the world, causing a global pandemic [2]. Now, COVID-19 continues to infect and kill individuals all over the world [3], As of 2022, COVID-19 has killed more than 6.5 million people, according to the WHO [4]. The COVID-19 pandemic, which has lasted for three years and is yet to end, has had a huge impact on the economy, politics, and many other parts of human society [5,6,7]. There has been a flurry of research on COVID-19 since 2020, while COVID-19 sequela is definitely a hot topic concentrated on by many scholars [8]. After observation of a large number of clinical cases, COVID-19 has been found to cause multi-organ sequelae [9], common sequelae include fatigue, headache, attention problems, hair loss and difficulty breathing [10]. At the same time, survivors of COVID-19 may also have anxiety, depression [11] and other mental problems as well as nervous system problems [12]. In a large number of studies on COVID-19 sequelae, some scholars have demonstrated the impact of COVID-19 sequelae by observing a large number of recovered patients over a long period of time and conducting cohort studies, and the sequelae of COVID-19 were found to be the result of multi-system involvement, including fatigue, loss of smell, cognitive dysfunction, and so on [13,14]. Among them, a prospective observational cohort study based on the first wave of the German epidemic published in 2022 found that many post-COVID-19 syndrome patients presented with symptoms of chronic fatigue syndrome [15].

ME/CFS is a systemic illness characterized by chronic and recurrent tiredness, which is frequently accompanied by anxiety, irritable bowel syndrome, fever, headache, muscular aches, and other symptoms [16]. A highly controversial condition in terms of both its existence and treatment [17], ME/CFS is a medically unexplained exhaustion that lasts for more than six months and is severe enough to cause a considerable decline in work, family, social, or school activities [18]. Many people in contemporary society are in a sub-healthy condition of chronic tiredness, and the occurrence of ME/CFS, as a widespread disease endangering human health, is rising year by year [19]. Long COVID is a chronic set of symptoms that patients may experience long after COVID remission. Clinical studies have reported that the range of symptoms in Long COVID patients, particularly fatigue, reduced daily activity and post-exercise discomfort, are very similar to those of ME/CFS [20]. However, although the symptoms of ME/CFS are similar to those of Long COVID, it remains to be verified whether ME/CFS is a sequel to COVID-19. Since the number of COVID-infected patients is increasing worldwide, exploring the relationship between COVID and ME/CFS is crucial for the later recovery of patients. This article focused on providing evidence for the link between COVID and ME/CFS.

We carried out a two-sample Mendelian randomization analysis to examine whether COVID-19 has a causative relationship with ME/CFS. The two-sample MR method eliminates the impact of reverse causality and confounding variables, which can skew the interpretation of traditional observational research. Finally, we discovered that there is no link between COVID-19 and ME/CFS.

## 2. Methods

The causal relationship regarding COVID-19 and ME/CFS is limited by traditional observational epidemiology and is susceptible to many confounding factors. Mendelian randomization (MR) is an important method for causal inference in epidemiology [21]. MR adopts genetic variation as an instrumental variable and it can overcome the shortcomings of traditional observational epidemiological studies such as poor extrapolation of results and difficulties in data acquisition [22]. Hence, in this study, we analyze the genome wide association study (GWAS) data by a two-sample MR approach [23], in order to examine whether there is a causal link between COVID-19 and ME/CFS. The two-sample MR (TSMR) analysis technique was employed to perform causal association analysis, before sensitivity analysis was undertaken to ensure the reliability of the results.

### 2.1. Data Sources and Processing

The COVID-19 Host Genetics Initiative provided us with GWAS summary information on COVID-19 severity, hospitalization, and susceptibility [24,25]. COVID-19 infection is defined as SARS CoV-2 infection identified by RT-PCR or patient self-reported infection. The data of ME/CFS was obtained from a study in UK bio bank [26], with N_case_ = 2076 and N_control_ = 460,857. Hence, excluding UK bio bank(UKBB), we selected the sets of GWAS summary statistics that did not contain the UKBB sample, in order to minimize the chance of sample overlap with GWAS data of ME/CFS. Susceptibility was examined between COVID-19 patients and COVID-19-free population controls, while the hospitalization phenotype was compared between patients with COVID-19 who were hospitalized and controls who were not admitted to hospitals because of COVID-19 or were COVID-19-free, severity phenotype was determined by comparing hospitalized COVID-19 patients who died or required respiratory assistance to controls who did not have severe COVID-19 or were free of COVID-19 [27]. Eventually, we get the following sample set, susceptibility: N_case_ = 143,839, N_control_ = 2,357,647; hospitalization: N_case_ = 40,929, N_control_ = 1,924,400; severity: N_case_ =17,472, N_control_ =725,695.

We used SNP as the instrumental variable, COVID-19 as the exposure variable, and ME/CFS as the outcome variable.

### 2.2. Selection of the Genetic IVs

Our selection of genetic IVs for performing TSMR should satisfy the following assumptions [28]: (1) there is a strong association between IVs and the exposure variable COVID-19; (2) IVs are not associated with any confounding factors related to the exposure variable COVID-19 and the outcome variable ME/CFS; and (3) IVs do not affect ME/CFS through any other pathways except those associated with the relation with exposure variable COVID-19.

In order to exclude the interference of strong linkage disequilibrium (LD) brought by SNPs, we specified the following screening settings for SNPs [29]: (1) with reference to the genomes of thousands of European people, we selected SNP with significant genome-wide significance with COVID (*p* < 5 × 10^−8^); and (2) the genetic distance between each two genes is at least 10,000 kb; (3) Set the r2 threshold for LD between genes to 0.001.

To evaluate the IVs, we also adopted Fstatistics [30]. If F > 10 then there is no weak instrumental variable bias, the statistics F are calculated as follows:(1)$$F=\frac{R^{2}}{1-R^{2}}\times \frac{N-M-1}{M}$$
where N denotes the exposure database’s sample size, M indicates the number of chosen SNPs, and R refers to the share of all variations explained by SNPs in the exposure dataset.
(2)$$R^{2}=\frac{2\times \left(1-\mathrm{MAF}\right)\times \mathrm{MAF}\times \beta }{\mathrm{SE}\times \sqrt{N}}$$

Here, MAF refers to minor allele frequency and β is the effect size of the SNPs on the exposed allele. MAF is equivalent to effect allele frequency (EAF) when computation. SE is the standard error of β. We can obtain these parameters directly from the selected SNPs.

### 2.3. TSMR Analysis

In this paper, inverse variance weighted (IVW) MR was used as the primary analysis method [28].

The concept of TSMR model is summarized in Figure 1. The IVW theorem holds that the fit is calculated by weighing the reciprocal of each result variance while guaranteeing that all IVs are valid. The IVW regression does not take into account the presence of the intercept term [31], whereas the MR-Egger regression includes the presence of the intercept term. The final result of IVW is a weighted average of the effect values of all instrumental variables, and when each genetic variant satisfies the IV hypothesis, IVW combines the Wald ratio estimates of the causal effects of different SNPs to provide a consistent estimate of the causal effect of exposure on outcome [32]. The weighted median method (WME) is defined as the weighted estimate of the ratio the median of the empirical density function, it provides the best estimate of the causal effect when at least half of the SNPs are valid IVs [33]. The MR-Egger method considers the presence of an intercept term when performing a weighted regression in the presence of multiplicity of instrumental variables and uses the intercept term to assess the magnitude of multiplicity among instrumental variables, and the slope is an estimate of the causal effect [34]. Simple mode is a simple estimation based on mode, which can be understood as the weighted median method with the same weight. However, when the estimation accuracy corresponding to different genetic variations is very different, this method has low efficiency [35]. When at least half of SNPs are valid, the weighted median method and weighted mode estimation can be used to obtain the estimation consistent with the final effect [36].

### 2.4. Sensitivity Analysis

In this paper, sensitivity analyses were conducted using other four TSMR methods based on other model assumptions to ensure the robustness of the results, and the other four methods were: weighted median estimator, simple median, MR-Egger regression, and weighted mode to conduct the relationship between exposure and outcome when the TSMR performed by all methods were statistically significant. Causality was robust. 

In addition to using different TSMR methods, we also performed sensitivity analyses such as the heterogeneity test and horizontal multiplicity test to ensure the robustness of the results. The heterogeneity test mainly reflects the difference between IVs, and the larger the difference between IVs, the greater the heterogeneity. Then this study used random effects to estimate the effect size of the MR. Cochran’s Q test and funnel plot were used to test for inter-IV heterogeneity. The pleiotropy test is used to test whether there is horizontal pleiotropy in multiple IVs, and the intercept term of the MR-Egger method is often used to indicate that if the difference between the intercept term and 0 is large, then there is horizontal pleiotropy. We also adopted mendelian randomization pleiotropy residual sum and outlier (MR-PRESSO) as a robustness check. MR-PRESSO removes abnormal SNPs (outliers) and estimates the corrected result, which avoids horizontal pleiotropy [37]. Additionally, ‘the leave one out’ sensitivity test, which is mainly used to eliminate IV one by one and then conduct TSMR analysis based on the remaining IVs to obtain the results, was also conducted for sensitivity analysis.

All the above analyses were done using the TwoSampleMR package [29] in R software version 4.2.1, and MR-PRESSO was done with the R package MRPRESSO. The evaluation indexes were the odds ratio (OR) and 95% confidence interval (95% CI). The differences were statistically significant when *p* < 0.05.

## 3. Results

### 3.1. SNPs

After removing the IVs with linkage disequilibrium, 57 SNPs were obtained in this paper. The specific details of IVs used in TSMR analysis in terms of severity, susceptibility and hospitalization of COVID-19 are given in Table 1.

### 3.2. TSMR Results

We analyzed the role of COVID-19 in the risk of ME/CFS by TSMR method. The results showed that COVID-19 severity, hospitalization, and susceptibility were not significantly associated with a higher risk of ME/CFS.

The results of TSMR analysis for all six methods are displayed in the forest plot in Figure 2. As is illustrated in Figure 2, no causal relationship between COVID-19 and ME/CFS was obtained for all five methods. The IVW results of TSMR analysis for COVID severity, hospitalization, and susceptibility were: severity (OR: 1.000, 95% CI: 0.999–1.000, *p* = 0.333); hospitalization (OR: 1.000, 95% CI: 0.999–1.001, *p* = 0.862); and susceptibility (OR: 1.000, 95% CI: 0.998–1.002, *p* = 0.998).

Table 2 also gives the specific results of all TSMR methods. In Table 2, β represents for the regression coefficient, SE means standard errors.

The scatter plot in Figure 3 shows the direction of the causal effect, and it is still not significant.

### 3.3. Sensitivity Analysis

To ensure the robustness of the TSMR results, we conducted a series of sensitivity analyses.

We used IVW and MR-Egger’s Cochran’s Q test to examine the heterogeneity of the individual causal effects. The results are shown in Table 3, and the *p*-values are not significant indicating that SNPs are not heterogeneous. Also the MR-Egger egger-intercept were not significantly statistical differences (all *p* values were greater than 0.05), so we can assume that SNPs have no horizontal pleiotropy. Again, none of the MR-PRESSO results were significant, showing that there was no horizontal pleiotropy.

The funnel plot in Figure 4 reveals that when a single SNP is used as the IV, the points generating the causal association effect are largely symmetrically distributed, indicating that the causal association is less likely to be affected by potential bias.

The results of the “Leave-one-out” sensitivity analysis are shown in Figure 5, in order of severity, hospitalization, and susceptibility. The results showed that after removing each SNP in turn, the IVW results for the remaining SNPs were not significantly different from the results for all SNPs. After removing SNPs one by one, the overall error line of the results did not change much, and the confidence intervals did not change much, so the removal of each SNP did not affect the results, indicating that the TSMR analysis was robust.

## 4. Discussion

In the previous study, we demonstrated by TSMR analysis that COVID-19 does not increase the risk of developing ME/CFS, and the results remained stable under a series of sensitivity analyses. However, the conclusions we obtained are not consistent with some clinical studies. As the number of COVID-19 infections continues to rise, there is widespread interest in the recovery of patients after a negative viral test, which scholars believe does not mean recovery; this phenomenon is known as “post-COVID” syndrome [38]. The latest clinical studies regard ME/CFS as one of the sequelae of COVID [15]. However, ME/CFS is a disease whose pathology has not been fully investigated [39], and symptoms after getting COVID-19 and symptoms after recovery from COVID-19, such as persistent muscle soreness, are very similar to those of ME/CFS and therefore may lead to confusion [20]. However, as a result, COVID-19 infection and ME/CFS may have mechanistic similarities, so it is also of clinical interest to study the two together [40]. In the two sample MR studies conducted above, we did not obtain evidence that COVID-19 causes ME/CFS. Therefore, we believe that clinically observed patients who have COVID-19 produce persistent pain and fatigue after getting COVID-19 probably do not have ME/CFS, but have symptoms similar to those of ME/CFS [41]. Therefore, randomized controlled trials on the sequelae of COVID-19 can consider other diseases.

However, although this paper concludes that ME/CFS is not a consequence of COVID sequelae by TSMR, we cannot arbitrarily assume that COVID-19 is not related to ME/CFS. This is still a controversial question and needs further research to provide an exact answer. The clinical features between Long COVID and ME/CFS are highly similar, and both include persistent fatigue, sleep problems, muscle aches, cognitive dysfunction and post-exercise discomfort, and in an observational trial prior to the COVID-19 pandemic, Long COVID and ME/CFS patients showed the same biological characteristics of these symptoms [42]. It is because of the high degree of symptom similarity between post-COVID and ME/CFS that a causal relationship between them has been sought by scholars. Until now, there is no exact pathophysiological explanation for COVID causing ME/CFS, it has been suggested that SARS-COV-2 may be the same physiological source of irritation as the causative agent of ME/CFS, which can cause ME/CFS-like symptoms in humans through the regulation of the hypothalamic paraventricular nucleus (PVN), called Post-COVID-19 Fatigue Syndrome [43]. A specific phenotype of ME/CFS is known as post-infection fatigue syndrome, which is associated with acute infection with viruses such as EBV [44]. The pathogenesis of both Long COVID and this type of ME/CFS is related to immune system dysregulation and high inflammatory response, etc., [20]. Therefore, there are great similarities between Long COVID and ME/CFS, and even if ME/CFS is not a sequela of COVID-19, the pathogenesis of ME/CFS and Long COVID may be similar, which means that the treatment of ME/CFS is likely to be useful for Long COVID, so it is important to continue to study the similarities and connections between COVID-19 and ME/CFS. 

In fact, we do not have definite evidence for the relationship between COVID-19 and ME/CFS, where errors and missing data are also important issues. For example, the data in this paper include some non-cancer illness code self-report ME/CFS cases, which may have led to data bias, and the estimated results may be wrong. In addition, the absence of micro-individual genetic data also leads to the conclusion that COVID is not causally related to ME/CFS when we cannot explore this issue in terms of gene expression. In summary, the estimation results of this paper using the IVW, MR-Egger regression method, weighted median, simple model, and weighted model are consistent, and the TSMR results do not suggest that COVID-19 and ME/CFS are causally related. Although we have not fully demonstrated whether there is an exact causal relationship between COVID-19 and ME/CFS, this paper has long been of interest because we provide evidence that ME/CFS is not a sequela of COVID-19.

However, there are many shortcomings in this study: (1) first, the sample population of this study is from Europe, and further studies are needed to verify whether the same conclusions can be drawn for other populations; (2) since this paper does not use individual-level data, it may not be adaptable to individual COVID patients. As this study uses a database of genetic variants without using specific microdata, the accuracy of the results cannot be guaranteed; (3) the results of this study are based on statistics and are not explained by biological mechanisms; (4) since this study does not contain a clinical trial, information regarding gene expression is also warranted to adjust for epigenetic biases; and (5)ME/CFS is clinically heterogeneous and may have a sex bias. Gender is likely to lead to heterogeneity, and gender differences in the relationship between COVID and ME/CFS will be worth discussing in future research.

## 5. Conclusions

In conclusion, this paper adopted two-sample mendelian randomization to demonstrate that COVID-19 is not causally related to ME/CFS, i.e., ME/CFS is not a sequela of COVID-19, but the specific relationship between COVID-19 and ME/CFS needs to be further investigated

## Figures and Tables

**Figure 1 ijerph-20-02437-f001:**
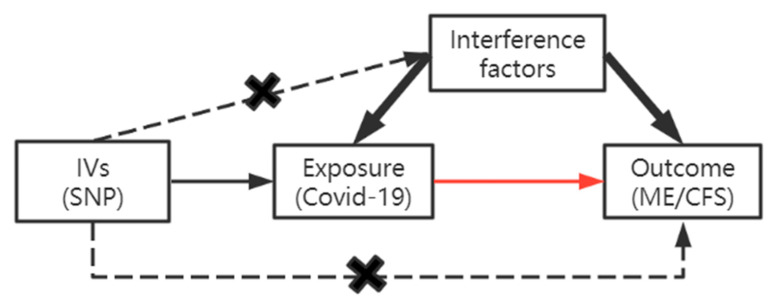
TSMR model.

**Figure 2 ijerph-20-02437-f002:**
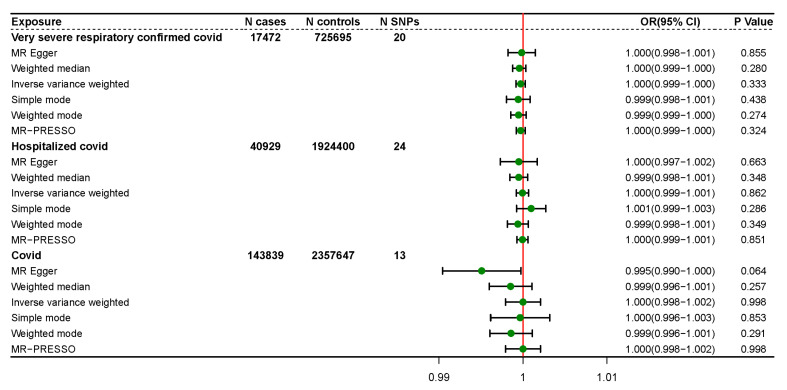
Forest Plots of TSMR Results.

**Figure 3 ijerph-20-02437-f003:**
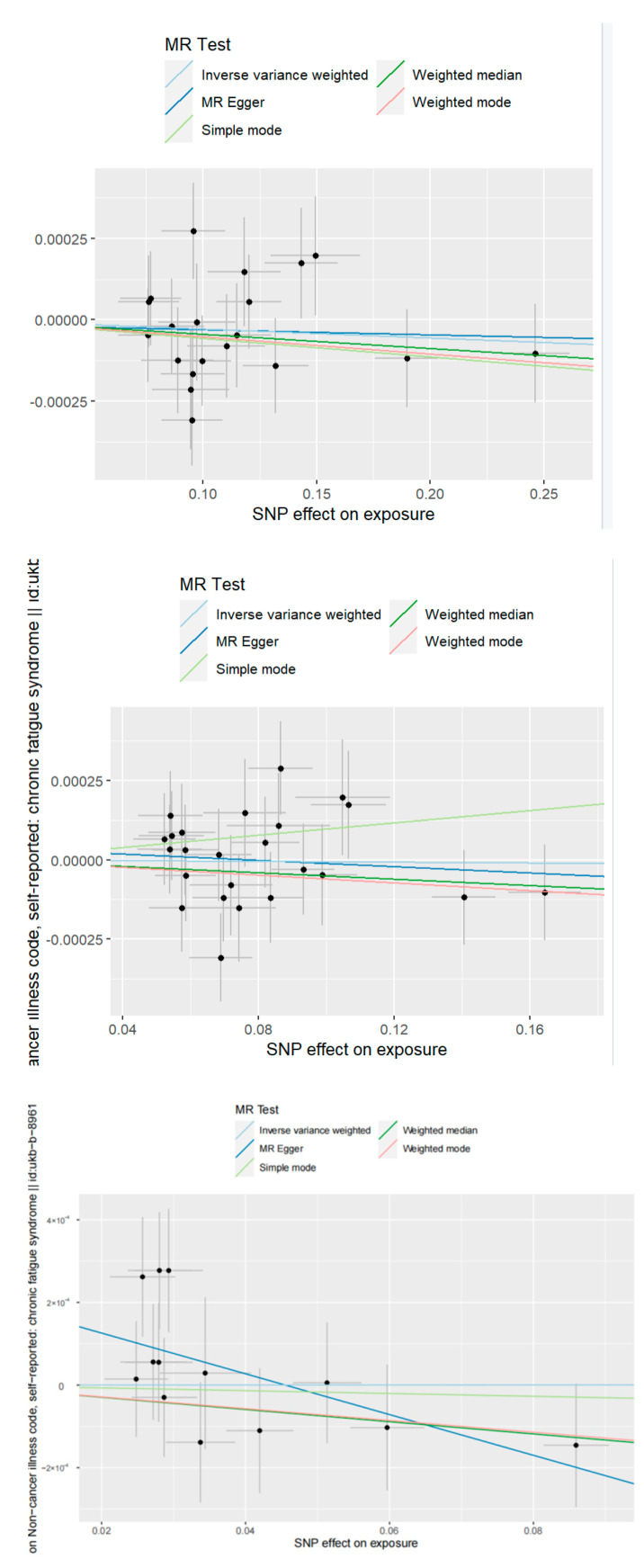
Scatter plots of TSMR Results: severity, hospitalization, and susceptibility.

**Figure 4 ijerph-20-02437-f004:**
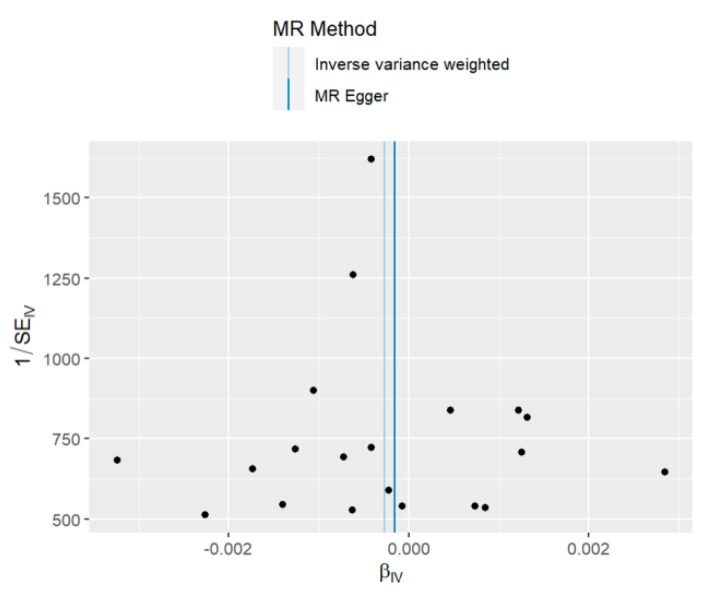
Funnel plots of TSMR Results: severity, hospitalization, and susceptibility.

**Figure 5 ijerph-20-02437-f005:**
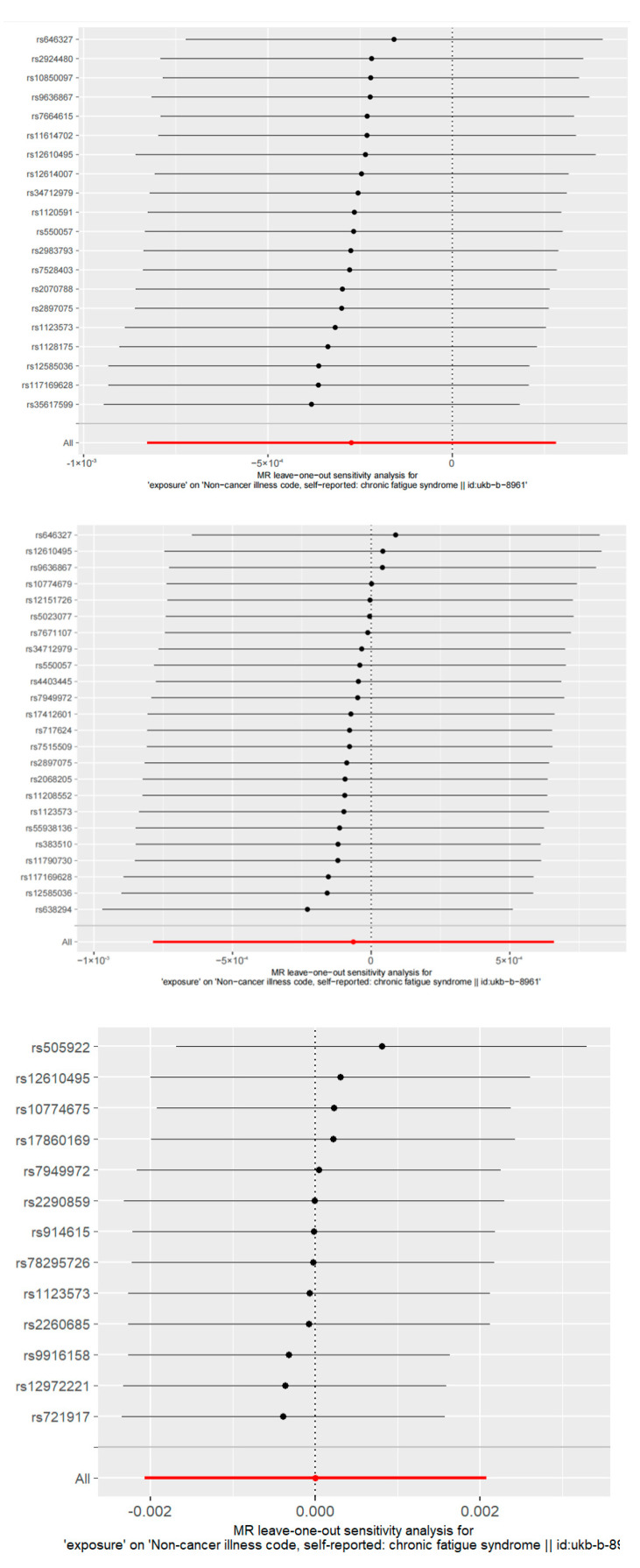
Sensitivity analysis results: “Leave-one-out”.

**Table 1 ijerph-20-02437-t001:** Detail Information for SNPs.

	SNP	Chromosome	Position	β.exposure	SE.exposure	pval.exposure	β.outcome	SE.outcome	pval.outcome	F-Value
**Severity**										
	rs10850097	12	113361117	0.09552	0.01399	8.59 × 10^−12^	−0.0001661	0.000145037	0.25	2907.13
	rs1120591	8	61537523	0.07600	0.01334	1.22 × 10^−8^	−0.0000482	0.000143872	0.74	2093.94
	rs1123573	2	60707588	−0.12036	0.01438	5.64 × 10^−17^	−0.0000551	0.000143234	0.70	4881.36
	rs1128175	6	31150435	−0.11827	0.01601	1.50 × 10^−13^	−0.0001474	0.000166854	0.38	4038.83
	rs11614702	12	133058157	0.09967	0.01306	2.29 × 10^−14^	−0.0001258	0.000138689	0.36	3704.18
	rs117169628	16	89264460	0.14967	0.01965	2.60 × 10^−14^	0.0001962	0.000183068	0.28	3629.32
	rs12585036	13	113535741	0.14326	0.01609	5.28 × 10^−19^	0.0001735	0.000170713	0.31	5210.72
	rs12610495	19	4717672	0.24613	0.01530	3.28 × 10^−58^	−0.0001029	0.000151855	0.50	19,886.87
	rs12614007	2	57316503	0.08889	0.01604	2.99 × 10^−8^	−0.0001245	0.000162446	0.44	2191.49
	rs2070788	21	42841988	−0.07615	0.01336	1.19 × 10^−8^	−0.0000560	0.000140635	0.69	2102.92
	rs2897075	7	99630342	0.07714	0.01334	7.30 × 10^−9^	0.0000651	0.000143991	0.65	2090.65
	rs2924480	11	34529831	−0.13204	0.01445	6.52 × 10^−10^	0.0001407	0.000146439	0.34	5875.94
	rs2983793	10	81445802	0.08640	0.01421	1.18 × 10^−9^	−0.0000194	0.000146350	0.89	2729.57
	rs34712979	4	106819053	−0.11042	0.01712	1.13 × 10^−10^	0.0000806	0.000159248	0.61	3370.15
	rs35617599	19	50874794	0.09594	0.01398	6.77 × 10^−12^	0.0002727	0.000148087	0.07	2960.59
	rs550057	9	136146597	0.11508	0.01501	1.73 × 10^−14^	−0.0000480	0.000159173	0.76	3956.96
	rs646327	19	49209851	−0.09538	0.01338	1.00 × 10^−12^	0.0003092	0.000139088	0.03	3310.72
	rs7528403	1	65382792	−0.09740	0.01686	7.58 × 10^−9^	0.0000079	0.000179455	0.96	2719.92
	rs7664615	4	25448493	−0.09472	0.01697	2.40 × 10^−8^	0.0002145	0.000184190	0.24	2142.10
	rs9636867	21	34609944	0.18972	0.01387	1.32 × 10^−42^	−0.0001185	0.000150267	0.43	12,505.91
**Hospitalization**										
	rs10774679	12	113374748	0.08358	0.00950	1.36 × 10^−18^	−0.0001198	0.000143988	0.41	6191.24
	rs11208552	1	65412830	−0.05736	0.00989	6.70 × 10^−9^	−0.0000870	0.000152264	0.57	3036.07
	rs1123573	2	60707588	−0.08188	0.01016	7.45 × 10^−16^	−0.0000551	0.000143234	0.70	6086.70
	rs117169628	16	89264460	0.10477	0.01391	5.09 × 10^−14^	0.0001962	0.000183068	0.28	4917.59
	rs11790730	9	33425871	0.07587	0.01218	4.68 × 10^−10^	0.0001500	0.000170557	0.38	3468.55
	rs12151726	2	198273591	0.05732	0.00975	4.11 × 10^−9^	−0.0001513	0.000139912	0.28	3094.13
	rs12585036	13	113535741	0.10643	0.01108	7.75 × 10^−22^	0.0001735	0.000170713	0.31	7364.79
	rs12610495	19	4717672	0.16432	0.01075	9.89 × 10^−53^	−0.0001029	0.000151855	0.50	22,020.48
	rs17412601	3	101499275	−0.06815	0.00978	3.24 × 10^−12^	−0.0000153	0.000145432	0.92	4097.22
	rs2068205	6	33058583	0.05430	0.00965	1.83 × 10^−8^	0.0000754	0.000140680	0.59	2706.14
	rs2897075	7	99630342	0.05229	0.00929	1.82 × 10^−8^	0.0000651	0.000143991	0.65	2494.14
	rs34712979	4	106819053	−0.07190	0.01214	3.21 × 10^−9^	0.0000806	0.000159248	0.61	3597.65
	rs383510	21	42858367	−0.05397	0.00946	1.14 × 10^−8^	−0.0001404	0.000139530	0.31	2836.26
	rs4403445	8	61432007	0.05845	0.00903	9.82 × 10^−11^	−0.0000500	0.000143867	0.73	3229.73
	rs5023077	12	133141973	−0.06962	0.00911	2.11 × 10^−14^	0.0001197	0.000138878	0.39	4766.32
	rs550057	9	136146597	0.09880	0.01030	8.74 × 10^−22^	−0.0000480	0.000159173	0.76	7169.36
	rs55938136	17	43798360	−0.08588	0.01524	1.73 × 10^−8^	−0.0001077	0.000166166	0.52	4029.68
	rs638294	19	50863023	0.08651	0.00945	5.67 × 10^−20^	0.0002897	0.000148130	0.05	6525.99
	rs646327	19	49209851	−0.06890	0.00933	1.54 × 10^−13^	0.0003092	0.000139088	0.026	4664.76
	rs717624	7	22894487	−0.05379	0.00947	1.34 × 10^−8^	−0.0000330	0.000140778	0.81	2678.99
	rs7515509	1	77949123	0.05838	0.00966	1.52 × 10^−9^	0.0000306	0.000142052	0.83	3101.91
	rs7671107	4	25449225	−0.07412	0.01099	1.51 × 10^−11^	0.0001520	0.000170930	0.37	4359.11
	rs7949972	11	34502042	−0.09310	0.00919	3.94 × 10^−24^	0.0000298	0.000143289	0.84	7757.94
	rs9636867	21	34609944	0.14045	0.00940	1.83 × 10^−50^	−0.0001185	0.000150267	0.43	17,671.68
**COVID**										
	rs10774675	12	113361237	0.03373	0.00476	1.38 × 10^−12^	−0.0001386	0.000144861	0.34	1206.40
	rs1123573	2	60707588	−0.02794	0.00465	1.83 × 10^−9^	−0.0000551	0.000143234	0.70	903.94
	rs12610495	19	4717672	0.05967	0.00509	1.02 × 10^−30^	−0.0001029	0.000151855	0.50	3665.96
	rs12972221	19	50879140	0.02932	0.00467	3.30 × 10^−10^	0.0002772	0.000148189	0.06	921.76
	rs17860169	21	34613301	0.04196	0.00461	8.90 × 10^−20^	−0.0001103	0.000150124	0.46	1984.44
	rs2260685	3	195497743	0.02716	0.00455	2.33 × 10^−9^	0.0000559	0.000139376	0.69	918.04
	rs2290859	3	101525625	−0.05132	0.00473	1.79 × 10^−27^	−0.0000054	0.000145542	0.97	2961.04
	rs505922	9	136149229	0.08592	0.00449	1.05 × 10^−81^	−0.0001457	0.000148905	0.33	8474.34
	rs721917	10	81706324	0.02802	0.00437	1.48 × 10^−10^	0.0002773	0.000140657	0.05	963.27
	rs78295726	19	10426512	0.03438	0.00629	4.60 × 10^−8^	0.0000290	0.000182971	0.87	727.61
	rs7949972	11	34502042	−0.02867	0.00450	1.87 × 10^−10^	0.0000298	0.000143289	0.84	937.44
	rs914615	1	155175892	−0.02480	0.00437	1.38 × 10^−8^	−0.0000145	0.000138969	0.92	761.27
	rs9916158	17	38182229	0.02569	0.00449	1.10 × 10^−8^	0.0002619	0.000143728	0.07	763.84

**Table 2 ijerph-20-02437-t002:** TSMR Results.

Exposure	Method	SNP	β	SE	*p*-Value	OR	OR (Lower 95% CI)	OR (Upper 95% CI)
Severity	MR Egger	20	−0.00015446	0.000832451	0.85487	0.99985	0.99822	1.00148
	Weighted median	20	−0.000436505	0.000403932	0.27986	0.99956	0.99877	1.00036
	IVW	20	−0.000273796	0.000282557	0.33255	0.99973	0.99917	1.00028
	Simple mode	20	−0.00056838	0.000716877	0.43765	0.99943	0.99803	1.00084
	Weighted mode	20	−0.00052857	0.000468997	0.27377	0.99947	0.99855	1.00039
	MR-PRESSO	20	−0.000273796	0.0002706	0.32436	0.99972	0.99919	1.00025
Hospitalization	MR Egger	24	−0.000495398	0.001123173	0.66347	0.99950	0.99731	1.00171
	Weighted median	24	−0.000506027	0.000538839	0.34768	0.99949	0.99844	1.00055
	IVW	24	−6.43 × 10^−5^	0.000368448	0.86153	0.99994	0.99921	1.00066
	Simple mode	24	0.000972112	0.000889543	0.28579	1.00097	0.99923	1.00272
	Weighted mode	24	−0.00060844	0.000636798	0.34928	0.99939	0.99815	1.00064
	MR-PRESSO	24	−6.38 × 10^−5^	0.0003366	0.85125	0.99993	0.99928	1.00059
Susceptibility	MR Egger	13	−0.004929688	0.002395122	0.06406	0.99508	0.99042	0.99976
	Weighted median	13	−0.001478204	0.001303262	0.25670	0.99852	0.99598	1.00108
	IVW	13	2.44 × 10^−6^	0.001058907	0.99817	1.00000	0.99793	1.00208
	Simple mode	13	−0.000339753	0.001793678	0.85293	0.99966	0.99615	1.00318
	Weighted mode	13	−0.001426332	0.001291833	0.29119	0.99857	0.99605	1.00111
	MR-PRESSO	13	2.44 × 10^−6^	1.0010589	0.99820	1.00000	0.99792	1.00208

As we can easily see from the table, no matter which approach is adopted, the *p*-value of the coefficient is higher than the 5% significant level, indicating that none of the results are statistically significant, hence there is no causal relationship between COVID-19 and ME/CFS.

**Table 3 ijerph-20-02437-t003:** Sensitivity analysis: heterogeneity, pleiotropy and MR-PRESSO.

Methods	Severity	Hospitalization	Susceptibility
**MR Egger-Intercept Pleiotropy Test *p*-value**	0.88	0.69	0.05
**MR Egger** Cochran’s Q **Test *p*-value**	0.50	0.64	0.63
**IVW** Cochran’s Q **Test *p*-value**	0.56	0.69	0.30
**MR-PRESSO Global Test *p*-value**	0.59	0.69	0.34

## Data Availability

The data are available under reasonable request to the corresponding author.

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
