# Peer review of "No Causal Effects Detected in COVID-19 and Myalgic Encephalomyelitis/Chronic Fatigue Syndrome: A Two Sample Mendelian Randomization Study"

_ijerph, 2023, doi:10.3390/ijerph20032437_

Round 1

Reviewer 1 Report

Very interesting study, bit still some questions occurred. ME/CFS is a disease whose pathology has not been fully investigated. How are you sure that there is absolutely no correlation between ME/CFS and COVID-19 ? It should be discussed more detailed.

And the latest clinical studies regard CFS as one of the sequels of Covid. However, ME/CFS is a disease whose pathology has not been fully investigatedsymptoms after getting Covid-19 and symptoms after recovery from Covid-19 such as persistent muscle soreness are very similar to those of ME/CFS and therefore may lead to confusion. The authors wrote this in the discussion, but I think it should be discussed in more detail, especially because post covid fatigue syndrome and classical CFS are almost equal. That's why I wrote, how authors can be 100 % sure that there is no correlation.However, as a result, Covid-19 infection and ME/CFS may have some similarities, so it is also of clinical interest to study the two together.

Author Response

Dear Reviewer:

Greetings,

Thank you so much for your comments and suggestions, here're the response for your comments (pdf version is pure while word version is trackable):

  1. As you said, the Pathogenesis of ME/CFS has not been fully investigated. Through this research, we conclude that there is no correlation between ME/CFS and COVID-19, however, without exact clinical evidence, we cannot draw the conclusion thoroughly, hence it should be discussed more detailed. Thus, we added a further discussion in the last part of the paper to discuss the reason why we suggest that there's no causal relationship between Covid and CFS.
  2. Also we discussed more in detail in the discussion part of the paper about the conclusion, we cannot 100% say that CFS is not a sequela of Covid, but in this paper, from the genetic data in GWAS, by using mendelian randomization approach, we suggest that Covid may not cause CFS in accordance with the results. Even though we can't 100% verify this conclusion, we give some evidence about this dispute, and I think that's the meaning of this paper, especially the Covid pandemic is still prevailing and more and more people may suffer from the long Covid, so it's crucial to figure out the relationship between CFS and those post Covid symptoms, this would be helpful for the recovering of patients after Covid. Though we don't fully confirm that the causal relationship between Covid and CFS, we give some genetic level evidence about this dispute and can be innovative for further studies, so we also added some new discussions in the drawbacks of this paper and suggestions for further research.

Thank you again for your professional review work on our article! With you all good in the new year!

Best regards.

Reviewer 2 Report

It is necessary to review the bibliography. Not all quotes follow the same structure. In some cases, identifying data is missing. 

Author Response

Dear reviewer:

Greetings,

Thank you so much for your comments on our article. According to your suggestions, we have supplemented several data here and corrected several mistakes in our previous draft. Based on your comments, we have made extensive revisions to our previous draft. The detailed point-by-point responses are listed below.

1.In introduction, we added more description of ME/CFS.

2.Structure errors and typo errors are addressed.

3. Some irrelevant cited references are deleted.

4. More discussion about the conclusion is given.

5.More details about the results are presented.

Thank you again for your kind help and wish you all good in the new year!

Best.

Reviewer 3 Report

The aim of the manuscript was to test for the causal association between COVID and ME/CFS, and to do that, the authors conducted two-sample mendelian randomization analysis using SNPs from GWAS studies as instrumental variables (IVs). Multiple genetic variants were deliberately selected for each COVID phenotype (susceptibility, hospitalization, and severity) to eliminate linkage disequilibrium (LD), and the estimates from each IV were combined using inverse variance weighted method similarly as in the meta-analysis literature. The authors also performed other MR methods to test for the robustness of IVW-MR, and sensitivity analyses were conducted to test for heterogeneity and the robustness of the selection of IVs. The author concluded that there was no causal relationship between COVID and ME/CFS, and that ME/CFS may not be a sequela of Covid-19. Upon review of the manuscript, I find that the study was appropriately designed, the statistical analysis was carefully conducted, the sample sizes were large enough for the proposed analysis, and the manuscript has great potential if the authors can address the following comments.

Major comments:

1.       ME/CFS is clinically heterogeneous and is highlighted by sex bias. The authors did test for heterogeneity, but only in the SNPs. Although sex may not be a confounder in the relationship (sex is not reported to be different in COVID infection), it may still act as an effect modifier due to sexual dimorphism. Therefore, I feel strongly that the authors should conduct sex stratification and examine the causal relationship in each sex, separately. In the Methods, lines 69-70, “…and gender heterogeneity was further examined using the Cochran Q test,…”, I don’t think the authors can say this until they conduct the sex stratification.

2.       Some of the inferences in Table 2 are puzzling. For example, the MR Egger model for COVID susceptibility produces an 95% CI not covering 0 (as seen in Fig. 2 and Table 2), however, the p-value is above 0.05. Some of the other p-values also did not comply with the Wald test principles, which makes me wonder if the authors added any random effects. If so, certain clarifications are imperatively needed; otherwise, the authors should check the accuracy of their data.

3.       Of the 5 models the authors performed, only the IVW-MR model was properly cited. The other 4 were not referenced, and I have a hard time understanding what “simple mode” and “weighted mode” models do exactly.

4.       Although the models were not referenced, it’s reasonable to assume that the 5 models were similar in structure, i.e., they all combined ratio estimates of single IVs. Then to rigorously test for the robustness of IVW-MR, it’s better to apply a model that’s fundamentally different. One possibility is to use an allele score with appropriate weights. Many studies have reported that MR analysis with an allele score produces robust estimates in causal inferences.

5.       The manuscript lacks the exploration of the SNP-SNP interactions. Additionally, I feel that given the complexity of ME/CFS, information on epigenetics (gene expression) is also warranted to adjust for epigenetic biases. Granted that the data is hard to collect, the authors should at least acknowledge this limitation adequately.

Minor comments:

1.       The authors used ME/CFS and CFS interchangeably. It’s better to be consistent and stick with ME/CFS.

2.       The background of ME/CFS needs to be covered with more details in the Introduction. The consistency in the symptoms between ME/CFS and long-COVID (PASC) should be mentioned.

3.       I’m not familiar with the UK ME/CFS biobank, and it is unclear how the ME/CFS cases were assessed. Were they all clinically diagnosed or were there any self-reported cases. If there were self-reported ME/CFS cases, this is a substantial limitation that needs to be addressed in the Discussion.

4.       Equation (2) is duplicated between lines 109 and 110.

5.       The F-statistics data of the SNP-exposure association are lacking. This information should be added to Table 1, instead of only presenting the p-values.

6.       The decimal places in Table 1 are not consistent, particularly, some p-values were presented in one or two decimal places, while others were much more precise. They need to be consistent.

7.       Table 2 can be better organized, it looks messy and confusing.1.        

8.       Line 204, it is unclear what the authors meant by “the results were on the right side of "0"”. Does it mean the confidence intervals all covered 0? Clarification is needed.

Author Response

Dear reviewer:

Greetings,

We feel great thanks for your professional review work on our article. As you are concerned, there are several problems that need to be addressed. According to your nice suggestions, we have made extensive corrections to our previous draft. We have added necessary data to supplement our results and edited our article extensively. The detailed corrections are listed below (Please note that pdf version is pure and word version is trackable):

Major comments:

  1. As you said, ME/CFS is clinically heterogeneous and is highlighted by gender bias. Conducting sex stratification may be a very meaningful work. However, after trying to do so, we find that we can’t get the gender information and relevant data divided by gender from GWAS, so it’s unlikely to do so in this study, we feel really regret and so sorry for that, we’ve acknowledged the shortcoming and presented in the last part of the paper. As we only test the heterogeneity in the SNPs, we’ve deleted the misleading sentence “gender heterogeneity was further examined using the Cochran Q test”.
  2. We’ve checked the data and the procedure again, we’ve pretty sure that we didn’t add any random effect, all codes are strictly adhered to the guidance of TSMR, and we used both R and Stata to conduct TSMR, finally we get the same results. As for the MR Egger model of susceptibility produces an 95% CI not covering 1 but the p-value is above 0.05.
  3. Sorry for our carelessness, as we adopt IVW as the core method, we ignored the description of the other 4 methods, now they’re referenced, and “simple mode” and “weighted mode” are further explained. 
  4. To ensure the robustness of IVW-MR, and in accordance with most of the MR papers (such as Li, C., Liu, J., Lin, J. et al. COVID-19 and risk of neurodegenerative disorders: A Mendelian randomization study. Transl Psychiatry 12, 283 (2022). https://doi.org/10.1038/s41398-022-02052-3), we added a different model MR-PRESSO as the robustness check. Using an allele score as IV may provide robust estimates in causal inferences, but when we’re doing so we still can’t obtain the allele score in the dataset we adopted in this article, this is really disappointing but we’ve learned a lot from your professional suggestions.
  5. Exploring SNP-SNP interactions and obtaining information on epigenetics need detailed individual level data, so the data is not available in GWAS, hence we admitted this limitation and also suggest other scholars to further investigate in the future.

We’ve tried our best to address the major comments, however, owing to data unavailability we cannot fix it perfectly, but of course we’re very willing to conduct further research according to the comments as we believe that it would be a great work. We have to admit that our work has certain limitations, however, most of the MR research also have such limitations (Such as the latest paper in a top journal: Mei J, Wei P, Zhang L, Ding H, Zhang W, Tang Y and Fang X (2023) Impact of ankylosing spondylitis on stroke limited to specific subtypes: Evidence from Mendelian randomization study. Front. Immunol doi:10.3389/fimmu.2022.1095622), the contribution of this paper is that, under the background of Covid pandemic, with increasing long Covid patients, we suggest that ME/CFS does not belong to long Covid by conducting a TSMR study, as they have so many similarities, we hope that it can help clinicians identify treatment between ME/CFS and long Covid more carefully, so as to promote the better recovery of long Covid patients. Though we can’t completely confirm the relationship of ME/CFS and Covid, we hope to provide evidence and with other scholar’s research, we believe that in one day the conclusion can be draw firmly.

Minor comments:

  1. We’ve replaced all CFS as ME/CFS.
  2. More information about ME/CFS is presented in the Introduction. And the similarity between ME/CFS and long-COVID is mentioned both in Introduction and Discussion.
  3. Some of the ME/CFS cases are non-cancer illness code self-reported, we’ve mentioned it in discussion part.
  4. The duplicated one is deleted. Thank you for reviewing the manuscript so careful, as we didn’t use the MDPI template when submitting, so the manuscript is re-edited by the publisher, and the original draft did not have those typo-errors, so really thank you for helping us find the mistakes!
  5. The F-statistics data of the SNP-exposure association are introduced to Table 1.
  6. The decimal places in Table 1 are now consistent.
  7. Thank you for pointing out the issue, to ensuring the table clearly reflect the results, we’ve deleted unnecessary items according to the existing MR papers, the CI of β. Also added some explanation in the context. See if it’s better now.
  8. Regarding “Leave-One-Out” test, more explanation of the results are given, and the misleading words are deleted.

Thank you again for you hard work and kind help for us, wish you all good in the new year!

Best regards.

Round 2

Reviewer 3 Report

The authors have addressed my concerns properly. Although some were left unaddressed due to data unavailability, corresponding limitations have been added in Discussion. I feel that the manuscript is ready for publication.